

# The association between the extracellular water-to-total body water ratio and albuminuria in Chinese type 2 diabetes mellitus patients

Aili Yang[1,*], Xinwen Yu[1,*], Zhiqiao Fan[2], Yuxin Jin[1], Fei Sun[1], Xin Wang[1], Xi Yuan[1], Langlang Liu[3], Guohong Zhao[1] and Bin Gao[1]

[1] Department of Endocrinology, Tangdu Hospital, Air Force Medical University, Xi'an, China
[2] Department of Hematology, Second Affiliated Hospital of Shaanxi University of Traditional Chinese Medicine, Xianyang, China
[3] Department of Endocrinology, Danfeng County Hospital, Shangluo, China
[*] These authors contributed equally to this work.

Corresponding authors
Guohong Zhao, zgh860204@163.com
Bin Gao, bingao0726@163.com

## ABSTRACT

**Background**. Diabetic kidney disease (DKD) is a common complication in patients with type 2 diabetes (T2DM), and early screening and diagnosis are crucial for preventing end-stage renal disease (ESRD). The extracellular water/total body water (ECW/TBW), as measured by bioelectrical impedance analysis (BIA), may be closely associated with the development of DKD. This study aimed to evaluate the relationship between ECW/TBW and albuminuria in T2DM patients and to explore its potential as an early diagnostic tool.

**Materials and methods**. This study included 1,034 T2DM patients. Demographic information, medical history, medication use, and laboratory test results were collected, including glycated hemoglobin (HbA1c), creatinine, lipid profile, and the urine albumin-creatinine ratio (UACR). BIA was used to measure parameters such as ECW/TBW. Multivariate logistic regression analysis explored the correlation between ECW/TBW and UACR. Ultimately, two simple nomograms were established to predict macroalbuminuria from patients with normoalbuminuria and microalbuminuria, respectively.

**Results**. The ECW/TBW increased significantly with rising UACR levels. Multivariate logistic regression analysis showed that ECW/TBW was significantly associated with macroalbuminuria compared to both normo-albuminuria and microalbuminuria (OR = 2.082, 95% CI [1.476–2.937], $P < 0.001$; and OR = 1.642, 95% CI [1.129–2.386], $P = 0.009$, respectively). In the analysis stratified by renal function, a similar relationship was found only in patients with eGFR $\geq$ 60 mL/min/1.73 m$^2$ (OR = 2.108, 95% CI [1.479–3.004], $P < 0.001$) but not in patients with eGFR < 60 mL/min/1.73 m$^2$. Finally, two nomograms for predicting macroalbuminuria were established. The C-index of the nomogram model for predicting the macroalbuminuria in patients with normoalbuminuria was 0.795 (95% CI [0.752–0.838]), and the C-index of the nomogram model for predicting the macroalbuminuria in patients with microalbuminuria was 0.761 (95% CI [0.711–0.812]).

**Conclusions**. This study demonstrated a significant correlation between the ECW/TBW and UACR levels in Chinese T2DM patients. In patients with normal or mildly

impaired renal function (eGFR $\geq$ 60 mL/min/1.73 m$^2$), ECW/TBW was significantly associated with macroalbuminuria, potentially serving as a diagnostic marker for macroalbuminuria.

## INTRODUCTION

Diabetes mellitus (DM) represents a global public health crisis, with escalating burdens worldwide. According to the International Diabetes Federation Diabetes Atlas (*Sun et al., 2022*), approximately 537 million adults had DM in 2021, projected to surge to 784 million by 2045. Over 90% of cases involve type 2 diabetes (T2DM), largely linked to modifiable risks. Critically, DM-related complications caused 6.7 million adult deaths in 2021, accounting for 12.2% of global all-cause mortality (*Cho et al., 2018*). Diabetic kidney disease (DKD), a predominant microvascular complication of diabetes, afflicts approximately 30–40% of individuals with diabetes (*Afkarian et al., 2016*; *Danta et al., 2021*). As the primary contributor to end-stage renal disease (ESRD) (*Cheng et al., 2021*), DKD exacerbates risks of cardiovascular morbidity and all-cause mortality (*Fox et al., 2012*), underscoring its dual burden on renal and systemic health.

DKD is typically asymptomatic in early stages, with insidious progression resulting in delayed diagnosis and intervention. Current clinical practice relies on periodic monitoring of urinary albumin-to-creatinine ratio (UACR) and estimated glomerular filtration rate (eGFR) as cornerstone diagnostic measures. Nevertheless, adherence to screening protocols remains suboptimal. A nationwide Chinese cross-sectional study involving 9,886 T2DM patients demonstrated a 32.5% prevalence of chronic kidney disease (CKD), yet only 26% of affected individuals were aware of their condition, with merely 55.3% receiving standardized screening (*Liu et al., 2022*). While emerging biomarkers derived from multi-omics approaches (genomics, proteomics, metabolomics) and advanced techniques like urinary exosome analysis show potential for early detection (*Thipsawat, 2021*), their clinical utility is hampered by operational complexity and limited accessibility. This underscores an urgent need for simplified, patient-centric warning systems to bridge the gap between novel biomarkers and real-world implementation.

Bioelectrical impedance analysis (BIA) is a valuable clinical tool for assessing body composition, including fat mass, muscle mass, and fluid distribution. BIA parameters are critical for evaluating metabolic health and guiding personalized interventions. Specifically, BIA-measured body fat percentage and visceral fat area correlate with hyperglycemia and dyslipidemia (*Sukkriang et al., 2021*), while muscle mass is inversely associated with blood glucose levels (*Sbrignadello, Gobl & Tura, 2022*). Additionally, BIA-derived visceral fat and body fat, combined with waist-to-hip ratio (WHR), enhance the prediction of metabolic-associated steatotic liver disease (MASLD) risk (*Song et al., 2025*). BIA also quantifies water content and distribution across body compartments, providing a comprehensive profile

of physiological status. The accuracy and repeatability of BIA had been validated (*Kyle et al., 2004*; *Ling et al., 2011*), and it is closely related to the gold-standard tracer dilution technique for assessing fluid status. BIA has been widely used in clinical practice to assess volume control in patients undergoing renal replacement therapy (*Davies & Davenport, 2014*). Recent studies have shown that increased extracellular fluid is significantly associated with decreased renal function in non-end-stage DKD patients (*Faucon et al., 2019*; *Low et al., 2021*), suggesting that extracellular fluid expansion is closely linked to the development of DKD. Albuminuria, a hallmark of DKD progression (*Raja, Maxwell & Brazil, 2021*) and an independent cardiovascular risk factor (*Cebrian et al., 2022*). A study by *Nakajima & Al (2021)* reported that, in a Japanese cohort of T2DM patients with eGFR ≥ 30 mL/min/1.73 m$^2$, an increased proportion of extracellular fluid was significantly associated with albuminuria, supporting the feasibility of predicting albuminuria by monitoring extracellular fluid proportion. However, few studies have specifically examined the relationship between extracellular fluid ratio and albuminuria in T2DM patients.

This study aims to investigate the association between the extracellular water ratio, as measured by BIA, and albuminuria levels in a Chinese T2DM population. Furthermore, it seeks to explore the relationship between these two parameters across different renal function stratifications based on eGFR, with the goal of identifying a rapid and effective method for screening patients with DKD.

## MATERIALS & METHODS

### Subjects

Between August 2022 and May 2024, patients with T2DM admitted to our institution were continuously screened. The inclusion criteria were as follows: (1) diagnosis of T2DM; (2) age ≥ 18 years; (3) underwent BIA for body fluid composition analysis; and (4) tested UACR. The exclusion criteria were as follows: (1) incomplete data, (2) presence of other kidney diseases, and (3) severe liver disease, (4) muscle disorders, (5) acute infection, (6) decompensated heart failure. This study was approved by the Institutional Review Board of the Medical Ethics Committee of Tangdu Hospital, Fourth Military Medical University, China (No. K202207-05), and complied with the principles of the Declaration of Helsinki. All participants provided written informed consent prior to participation in the study.

### Data collection

Demographic data, including the duration of T2DM, smoking history (defined as smoking at least one cigarette per day for at least six months), and alcohol consumption history (defined as consumption for at least one year), were collected. The medication history of the patients was recorded, including the use of antidiabetic drugs and antihypertensive drugs, such as insulin, metformin hydrochloride, sodium-glucose cotransporter 2 inhibitors (SGLT-2i), glucagon-like peptide 1 receptor agonists (GLP-1RA), renin–angiotensin–aldosterone system (RAAS) inhibitors, calcium channel blockers (CCB), and diuretics. Height and weight were measured by a trained nurse in the morning while the patients were fasting. Blood pressure was measured following a 10-minute rest period in a seated position. To ensure accuracy, blood pressure was measured at least twice for each participant, with

at least one-minute interval between measurements. The average of these two readings was then calculated and recorded as the participant's blood pressure for the study.

Laboratory indicators, including glycated hemoglobin (HbA1c), creatinine, and lipid profile including total cholesterol (TC), triglycerides (TG), low-density lipoprotein cholesterol (LDL-C), high-density lipoprotein cholesterol (HDL-C), fasting plasma glucose (FBG), potassium, sodium, hematocrit were collected, after an overnight fast. eGFR was calculated using the CKD-EPI formula (*Levey et al., 2009b*), eGFR = a × (Scr/b) c × $(0.993)^{age}$. a: female = 144, male = 141; b: female = 0.7; male = 0.9; c: female, Scr ≤ 0.7 mg/dl = −0.329, Scr > 0.7 mg/dl = −1.209; male, Scr ≤ 0.9 mg/dl = −0.411; Scr > 0.9 mg/dl = −1.209. TyG Index = ln (fasting triglycerides (mg/dL) × FBG (mg/dL)/2). Urine albumin and creatinine concentrations were measured from midstream urine after waking up, and the UACR was calculated. The participants were devided into three groups: normo-albuaminuria (UACR < 30 mg/g), microalbuaminuria (30–300 mg/g) an0d macroalbuaminuria (UACR > 300 mg/g).

Body composition parameters, including extracellular water (ECW), intracellular water (ICW), total body water (TBW), ECW/TBW ratio, body fat percentage, waist-to-hip ratio, and skeletal muscle mass, were measured using a bioelectrical impedance analyzer (BIA; Model: Biospace, Seoul, South Korea; Serial No.: 720). Prior to measurement, participants were required to fast for ≥8 h and abstain from water intake for ≥4 h. They were further instructed to avoid strenuous exercise, sauna use, and excessive fluid consumption within 24 h before testing. Standardized measurement protocols were followed: participants wore lightweight clothing, and women of reproductive age were advised to schedule assessments outside their menstrual periods.

### Statistical analysis

All statistical analyses were performed using SPSS version 26.0 (IBM Corp., Armonk, NY, USA). Continuous variables were expressed as means ± standard deviations, and categorical variables were expressed as proportions. One-way analysis of variance (ANOVA) was used for normally distributed continuous variables, while nonparametric tests were used for variables that were not normally distributed. Chi-square tests were used for categorical variables. Multivariate logistic regression analysis was employed to examine factors affecting UACR in T2DM patients. Finally, two predictive models for macroalbuminuria were constructed, nomograms were depicted and used to evaluate the prediction accuracy through ROC curve analysis, and were evaluated by the concordance index (C-index) and the calibration curve. $P$-value < 0.05 was considered statistically significant.

## RESULTS

### Baseline characteristics

A total of 1,034 patients with T2DM were enrolled in this study, the baseline characteristics were shown in Table 1. Among the participants, 640 (58.4%) had normo-albuminuria, 244 (23.6%) had microalbuminuria, and 150 (14.5%) had macroalbuminuria. Patients with microalbuminuria and macroalbuminuria had a longer duration of diabetes, were more often male, had higher blood pressure, more extracellular fluid, higher creatinine

**Table 1 Characteristics of study patients.**

| Variables | Total | Normo-albuminuria | Microalbuminuria | Macroalbuminuria | P |
|---|---|---|---|---|---|
| n (%) | 1,034 | 640 (61.90) | 244 (23.60) | 150 (14.50) | — |
| Age (years) | 54.40 ± 12.87 | 54.02 ± 12.76 | 55.48 ± 11.94 | 54.27 ± 14.66 | 0.315 |
| Male, n (%) | 722 (69.80) | 425 (66.40) | 182 (74.50) | 115 (76.70) | 0.009 |
| Duration of diabetes (years) | 8 (3, 14) | 8 (2, 13) | 8 (3, 14) | 10 (4, 18) | <0.001 |
| Smoking, n (%) | 359 (34.72) | 221 (34.53) | 86 (35.25) | 52 (34.67) | 0.980 |
| Systolic pressure (mmHg) | 130.71 ± 16.29 | 127.99 ± 14.37 | 131.23 ± 17.17 | 141.75 ± 17.97 | <0.001 |
| Diastolic pressure (mmHg) | 82.41 ± 10.09 | 81.33 ± 9.73 | 82.69 ± 10.00 | 86.66 ± 10.65 | <0.001 |
| BMI (kg/m$^2$) | 25.74 ± 3.71 | 25.56 ± 3.67 | 25.95 ± 3.51 | 26.15 ± 4.16 | 0.125 |
| Body composition | | | | | |
| Body fat mass (kg) | 23.30 ± 10.80 | 23.17 ± 10.54 | 24.01 ± 11.51 | 22.67 ± 10.67 | 0.436 |
| Body fat percentage (%) | 29.21 ± 7.84 | 29.40 ± 7.76 | 29.01 ± 7.61 | 28.68 ± 8.51 | 0.541 |
| Skeletal muscle mass (kg) | 27.58 ± 5.31 | 27.30 ± 5.35 | 27.88 ± 4.96 | 28.30 ± 5.62 | 0.07 |
| Waist-to-hip ratio | 0.92 ± 0.07 | 0.92 ± 0.07 | 0.93 ± 0.06 | 0.91 ± 0.08 | 0.059 |
| ICW (kg) | 22.74 ± 4.08 | 22.54 ± 4.09 | 23.02 ± 3.83 | 23.13 ± 4.36 | 0.13 |
| ECW (kg) | 14.34 ± 2.46 | 14.13 ± 2.42 | 14.48 ± 2.30 | 14.98 ± 2.77 | 0.001 |
| TBW (Kg) | 38.88 ± 9.34 | 38.65 ± 9.69 | 39.65 ± 9.21 | 38.60 ± 7.91 | 0.338 |
| ECW/TBW ratio (%) | 38.71 ± 1.19 | 38.58 ± 0.76 | 38.70 ± 0.88 | 39.27 ± 1.19 | <0.001 |
| HbA1c (%) | 8.59 ± 2.19 | 8.44 ± 2.19 | 8.67 ± 1.98 | 9.08 ± 2.46 | 0.005 |
| Creatinine (μmol/L) | 60 (51, 71) | 59 (50, 67) | 60 (49, 71) | 76 (56, 113) | <0.001 |
| eGFR (mL/min/1.73 m$^2$) | 102.79 ± 23.17 | 106.71 ± 16.85 | 103.15 ± 20.65 | 85.37 ± 37.98 | <0.001 |
| ≥60 mL/min/1.73 m$^2$, n (%) | 54 (5.20) | 6 (0.90) | 8 (3.30) | 40 (26.70) | <0.001 |
| ≥60 mL/min/1.73 m$^2$, n (%) | 980 (94.80) | 634 (99.10) | 236 (96.70) | 110 (73.30) | |
| Total protein (g/L) | 68.06 ± 7.02 | 67.57 ± 6.53 | 69.93 ± 7.03 | 67.18 ± 8.4 | <.0001 |
| Albumin (g/L) | 43.99 ± 5.45 | 44.09 ± 4.72 | 45.06 ± 5.79 | 41.81 ± 7.02 | <0.001 |
| TC (mmol/L) | 4.44 ± 1.21 | 4.33 ± 1.13 | 4.34 ± 1.07 | 5.10 ± 1.53 | <0.001 |
| TG (mmol/L) | 2.19 ± 2.02 | 2.14 ± 2.01 | 2.02 ± 1.37 | 2.65 ± 2.75 | 0.007 |
| LDL-C (mmol/L) | 2.48 ± 0.90 | 2.42 ± 0.87 | 2.44 ± 0.82 | 2.85 ± 1.06 | <0.001 |
| HDL-C (mmol/L) | 1.09 ± 0.27 | 1.08 ± 0.27 | 1.07 ± 0.27 | 1.16 ± 0.28 | 0.001 |
| FBG (mmol/L) | 8.40 ± 3.21 | 8.24 ± 3.14 | 8.55 ± 3.00 | 8.87 ± 3.75 | 0.072 |
| TyG Index | 9.29 ± 0.80 | 9.25 ± 0.82 | 9.30 ± 0.70 | 9.49 ± 0.85 | 0.004 |
| Potassium (mmol/L) | 4.18 ± 0.46 | 4.12 ± 0.41 | 4.22 ± 0.44 | 4.36 ± 0.62 | <.001 |
| Sodium (mmol/L) | 140.89 ± 38.74 | 141.75 ± 49.10 | 139.42 ± 2.20 | 139.67 ± 8.93 | 0.671 |
| Hematocrit (%) | 42.85 ± 5.09 | 42.86 ± 4.38 | 44.23 ± 4.98 | 40.69 ± 6.96 | <.001 |
| Hemoglobin (g/L) | 142.14 ± 19.74 | 142.35 ± 16.68 | 146.76 ± 22.14 | 134.11 ± 24.73 | <.001 |
| Use of insulin, n (%) | 427 (41.30) | 243 (38.00) | 101 (41.40) | 83 (55.30) | 0.001 |
| Use of SGLT-2 inhibitor, n (%) | 168 (16.20) | 101 (15.80) | 35 (14.30) | 32 (21.30) | 0.165 |
| Use of diuretics, n (%) | 56 (5.40) | 28 (4.40) | 13 (5.30) | 15 (10.00) | 0.023 |
| Use of RAAS inhibitor, n (%) | 211 (20.40) | 106 (16.60) | 54 (22.10) | 51 (34.00) | <0.001 |
| Use of CCB, n (%) | 261 (25.20) | 140 (21.90) | 59 (24.20) | 62 (42.30) | <0.001 |

**Notes.**

BMI, body mass index; ICW, intracellular water; ECW, extracellular water; TBW, total body water; HbA1c, glycated haemoglobin; eGFR, estimated glomerular filtration rate; TC, total cholesterol; TG, triglycerides; LDL-C, low-density lipoprotein cholesterol; HDL-C, high-density lipoprotein cholesterol; FBG, Fasting Plasma Glucose; TyG, triglyceride-glucose index; SGLT-2, sodium-dependent glucose transporters 2; RAAS, renin-angiotensin-aldosterone system; CCB, calcium channel blockers.

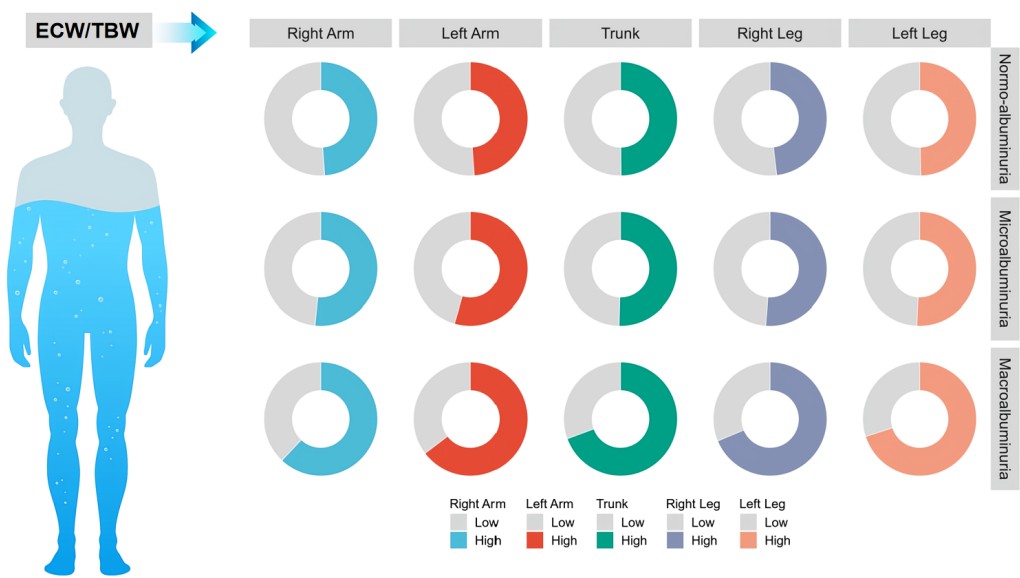

**Figure 1** **Comparison of the ECW/TBW ratio across different body regions among three participant groups.** Patients were categorized based on the median ECW/TBW ratios of various body regions, and the differences in ECW/TBW across distinct UACR groups were expressed as percentages.

levels, lower eGFR, higher lipid levels, and a higher proportion use insulin, diuretics and antihypertensive medications ($P < 0.05$) (Table 1).

## Comparative analysis of ECW/TBW across various anatomical regions among three UACR groups

To further investigate whether the variations in ECW/TBW among different albuminuria groups were associated with specific anatomical regions, patients were categorized into two groups based on the median ECW/TBW for distinct body segments, including the left upper limb, right upper limb, trunk, and left lower limb, and the differences in ECW/TBW across distinct UACR groups were expressed as percentages. The results indicated that as UACR increased, the ECW/TBW across all anatomical regions exhibited a gradual increase (Fig. 1) (Table S1), suggesting that the relationship between ECW/TBW and UACR has no anatomical regional specificity.

## Analysis of the relationship between ECW/TBW and UACR groups

Multifactorial logistic regression models examined the association between ECW/TBW and UACR. In comparison to the normo-albuminuria group, ECW/TBW demonstrated a significant correlation with macroalbuminuria ($P < 0.001$), while no significant association was observed with microalbuminuria. Similarly, in comparison to microalbuminuria group, ECW/TBW remained significantly associated with macroalbuminuria ($P < 0.001$) (Table 2), indicating a robust association between ECW/TBW and macroalbuminuria. Given that no significant difference in ECW/TBW was observed between the normo-albuminuria and microalbuminuria groups, further analysis was conducted to elucidate the relationship between ECW/TBW and macroalbuminuria by dividing patients into

**Table 2    Association between ECW/TBW ratio and UACR.**

| | Unadjusted | | Model 1 | | Model 2 | | Model 3 | |
|---|---|---|---|---|---|---|---|---|
| | OR (95% CI ) | P | OR (95% CI ) | P | OR (95% CI ) | P | OR (95% CI ) | P |
| Normo-albuminuria | Ref. | | Ref. | | Ref. | | Ref. | |
| Microalbuminuria | 1.178 (0.989~1.404) | 0.067 | 1.215 (0.995~1.485) | 0.056 | 1.271 (0.998~1.618) | 0.052 | 1.268 (0.993~1.620) | 0.057 |
| Macroalbuminuria | 2.293 (1.866~2.817) | <0.001 | 2.647 (2.086~3.358) | <0.001 | 2.106 (1.509~2.938) | <0.001 | 2.082 (1.476~2.937) | <0.001 |
| Microalbuminuria | Ref. | | Ref. | | Ref. | | Ref. | |
| Macroalbuminuria | 1.946 (1.545~2.450) | <0.001 | 2.178 (1.673~2.835) | <0.001 | 1.657 (1.154~2.379) | 0.006 | 1.642 (1.129~2.386) | 0.009 |

Notes.

Model 1 adjusted for age, gender, duration of diabetes, BMI, and current smoker.

Model 2 adjusted for age, gender, duration of diabetes, BMI, Current smoker, Systolic pressure, Diastolic pressure, LDLC, HDLC, HbA1c, creatinine, TyG Index, Potassium, Albumin, Hematocrit.

Model 3 adjusted for age, gender, duration of diabetes, BMI, Current smoker, Systolic pressure, Diastolic pressure, LDLC, HDLC, HbA1c, creatinine, TyG Index, Potassium, Albumin, Hematocrit, use of Insulin, SGLT-2 inhibitor, diuretics, RAAS inhibitor, CCB.

**Table 3    Association between ECW/TBW ratio and UACR in different eGFR grades.**

| | Unadjusted | | Model 1 | | Model 2 | | Model 3 | |
|---|---|---|---|---|---|---|---|---|
| | OR (95% CI) | P | OR (95% CI) | P | OR (95% CI) | P | OR (95% CI) | P |
| Normo+Microalbuminuria | Ref. | | Ref. | | Ref. | | Ref. | |
| Macroalbuminuria | 2.182 (1.790~2.660) | <0.001 | 2.489 (1.981~3.127) | <0.001 | 2.047 (1.515~2.766) | <0.001 | 2.037 (1.485~2.795) | <0.001 |
| eGFR < 60 mL/min/1.73 m$^2$ | | | | | | | | |
| Normo+Microalbuminuria | Ref. | | Ref. | | Ref. | | Ref. | |
| Macroalbuminuria | 1.595 (0.903~2.820) | 0.108 | 1.741 (0.906~3.344) | 0.096 | 1.607 (0.506~5.100) | 0.421 | 1.369 (0.354~5.291) | 0.649 |
| eGFR≥60 mL/min/1.73 m$^2$ | | | | | | | | |
| Normo+Microalbuminuria | Ref. | | Ref. | | Ref. | | Ref. | |
| Macroalbuminuria | 1.770 (1.400~2.237) | <0.001 | 2.287 (1.754~2.982) | <0.001 | 2.126 (1.505~3.003) | <0.001 | 2.108 (1.479~3.004) | <0.001 |

Notes.

Model 1 adjusted for age, gender, duration of diabetes, BMI, and current smoker.

Model 2 adjusted for age, gender, duration of diabetes, BMI, Current smoker, Systolic pressure, Diastolic pressure, LDLC, HDLC, HbA1c, TyG Index, Potassium, Albumin, Hematocrit.

Model 3 adjusted for age, gender, duration of diabetes, BMI, Current smoker, Systolic pressure, Diastolic pressure, LDLC, HDLC, HbA1c, TyG Index, Potassium, Albumin, Hematocrit, use of Insulin, SGLT-2 inhibitor, diuretics, RAAS inhibitor, CCB.

two groups: one comprising those with normo-albuminuria plus microalbuminuria and another consisting of those with macroalbuminuria. Binary logistic regression models were employed to assess differences in ECW/TBW between these two cohorts; the results consistently revealed a significant association between ECW/TBW and macroalbuminuria ($P < 0.001$) (Table 3).

Furthermore, to investigate whether the association between ECW/TBW and macroalbuminuria differs among patients with varying grades of renal function, participants were categorized into two groups based on eGFR: one group with eGFR < 60 mL/min/1.73 m$^2$ and another with eGFR $\geq$ 60 mL/min/1.73 m$^2$. The relationship between ECW/TBW and macroalbuminuria was assessed in these two groups using binary logistic regression models. It was found that in patients with eGFR $\geq$ 60 mL/min/1.73 m$^2$, ECW/TBW demonstrated a significant association with macroalbuminuria ($P < 0.001$). Conversely, in patients with eGFR < 60 mL/min/1.73 m$^2$, no significant association between

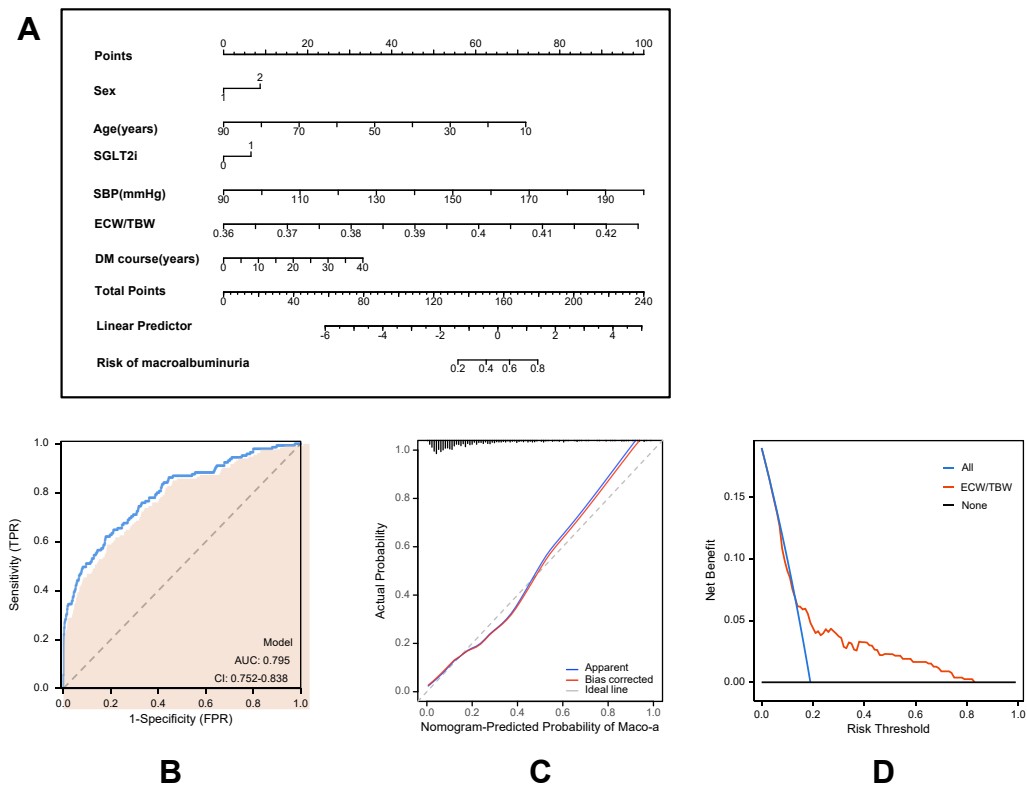

**Figure 2  Model for predicting macroalbuminuria in T2DM patients with normal albuminuria.** (A) Nomogram model for predicting the development of macroalbuminuria in T2DM patients with normal albuminuria. (B) Receiver operating characteristic (ROC) curve of the prediction model. AUC: Area under the curve. (C) Calibration curve of the nomogram model. (D) Decision curve analysis (DCA) of the prediction model.

ECW/TBW and macroalbuminuria was identified; this indicates that in T2DM patients with preserved renal function (eGFR grades 1–2), a significant association exists between ECW/TBW and macroalbuminuria (Table 3).

## Development and validation of an individualized prediction model
Based on multivariate logistic regression models, two predictive models were developed to estimate the risk of macroalbuminuria, targeting individuals with normoalbuminuria and microalbuminuria, respectively. Model A predicts the progression from normoalbuminuria to macroalbuminuria (Fig. 2), while Model B predicts the progression from microalbuminuria to macroalbuminuria (Fig. 3). Both models incorporated variables including age, sex, duration of type 2 diabetes, use of SGLT2 inhibitors, systolic blood pressure, and the ECW/TBW ratio. Corresponding nomograms were constructed for each model (Figs. 2A and 3A), where each variable's contribution to macroalbuminuria risk is represented by a specific point value. The total risk score was calculated by summing the points of all variables, providing an individualized estimate of macroalbuminuria risk in patients with type 2 diabetes.

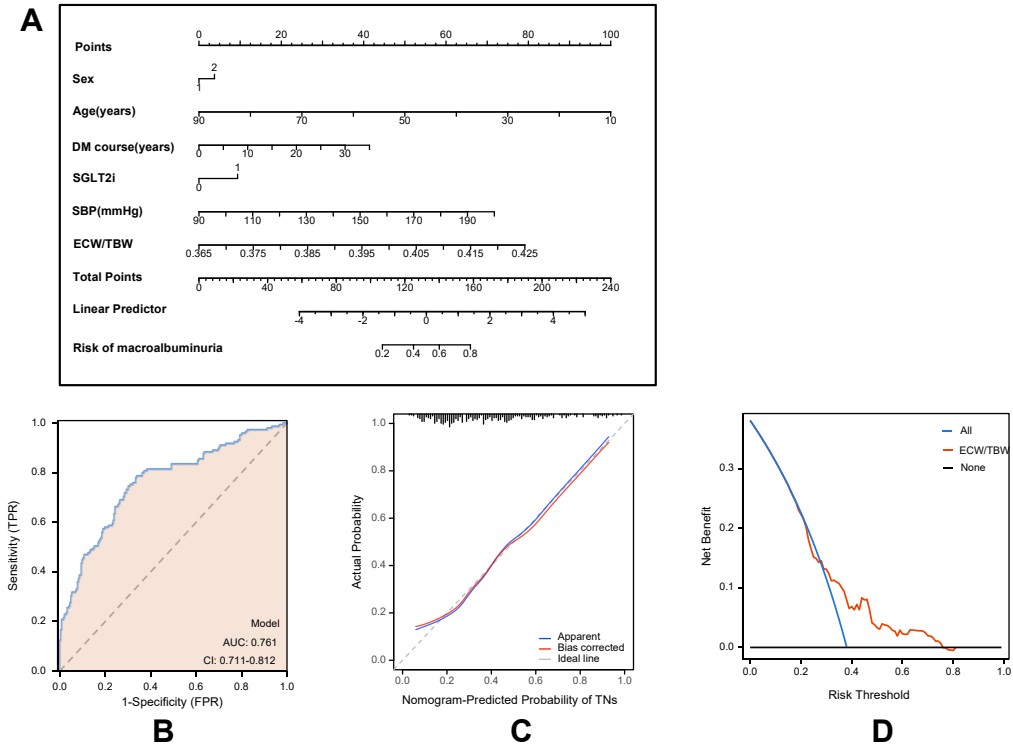

**Figure 3 Model for predicting macroalbuminuria in T2DM patients with microalbuminuria.**
(A) Nomogram model for predicting the development of macroalbuminuria in T2DM patients with microalbuminuria. (B) Receiver operating characteristic (ROC) curve of the prediction model. AUC: area under the curve. (C) Calibration curve of the nomogram model. (D) Decision curve analysis (DCA) of the prediction model.

Model performance was evaluated using receiver operating characteristic (ROC) curves, calibration curves, and decision curve analysis (DCA). ROC analysis demonstrated that Model A achieved an AUC of 0.795 (95% CI [0.752–0.838]; Fig. 2B), while Model B achieved an AUC of 0.761 (95% CI [0.711–0.812]; Fig. 3B), indicating robust predictive accuracy. Calibration curves showed excellent agreement between predicted probabilities and observed outcomes (Figs. 2C and 3C). Hosmer-Lemeshow tests confirmed good model calibration (Model A: $P = 0.469$; Model B: $P = 0.197$). DCA further evaluated the clinical utility of the nomograms, demonstrating high net clinical benefit for predicting macroalbuminuria across a threshold probability range of 0% to 80% (Figs. 2D and 3D). These findings underscore the strong predictive performance, clinical applicability, and decision-support value of the two nomogram models.

## DISCUSSION

Early diagnosis and intervention of DKD are critical for preventing ESRD and mitigating the risk of cardiovascular complications (*Coresh et al., 2019*; *Heerspink et al., 2019*). However, owing to the subtle early manifestations of DKD, many patients remain unscreened or undiagnosed promptly, resulting in delayed treatment. This study demonstrated that among

patients with T2DM, the ECW/TBW exhibited a strong correlation with the incidence of macroalbuminuria, particularly among those classified as eGFR grades 1 and 2. Suggesting that ECW/TBW has the potential to early identify macroalbuminuria in T2DM patients, which is essential for effective clinical management and improved prognosis.

This study further revealed that in T2DM patients, the ECW/TBW increased significantly with rising UACR, confirming the positive correlation between increased extracellular fluid ratio and albuminuria. A previous study by *Low et al. (2021)* explored the use of BIA to assess fluid status in DKD patients, showing that the proportion of extracellular water was closely related to renal function decline. *Nakajima & Al (2021)* focused on patients with eGFR grades 1–3, revealing a strong association between increased extracellular fluid and albuminuria, and proposed that the ECW/ICW ratio could serve as a potential indicator for predicting albuminuria. Our study extended these findings to include T2DM patients with eGFR grades 4–5 and replaced the ECW/ICW ratio with the ECW/TBW. Therefore, our study indicated that ECW/TBW and UACR levels demonstrate consistent trends across varying stages of renal function in patients with T2DM, suggesting the universal applicability of this indicator among T2DM patients with differing kidney function.

The observed increase in ECW/TBW may result from either an elevation in ECW or a reduction in ICW. In this study, both ECW and ICW were found to rise with increasing levels of albuminuria; however, the increase in ECW was more pronounced, resulting in a higher ECW/TBW. The causal relationship between albuminuria and increased ECW remains ambiguous, potentially linked to a reduction in plasma colloid osmotic pressure resulting from albumin loss. Our data indicate that patients with macroalbuminuria exhibit the lowest serum albumin levels. Furthermore, the development of albuminuria is influenced by multiple factors, including inflammation, oxidative stress, and hemodynamic disturbances. For instance, inflammation may enhance capillary permeability, leading to fluid redistribution. However, the predictive capacity of ECW/TBW was most pronounced in patients with CKD stages 1–2, while it significantly diminished in those with CKD stages 3–5. In advanced CKD, reduced eGFR and impaired water and sodium excretion likely play a more pivotal role in fluid imbalance, thereby attenuating the correlation between ECW/TBW and proteinuria. Consequently, our study not only corroborated *Nakajima & Al's (2021)* findings regarding the predictive value of extracellular fluid ratio in early-stage CKD patients but also emphasized the limitations of this indicator in CKD stages 3–5. This highlights the heterogeneity among patients at different CKD stages and underscores the significance of ECW/TBW for detecting macroalbuminuria in individuals with preserved renal function.

Although this study did not identify a significant predictive value for the ECW/TBW in detecting microalbuminuria, it demonstrated substantial clinical relevance in predicting macroalbuminuria. In contrast to microalbuminuria, macroalbuminuria is frequently an irreversible pathological condition (*Levey et al., 2009a*; *Levey et al., 2009b*; *Levey et al., 2020*) and markedly increases the risk of progression to ESRD (*Tsai et al., 2015*). While microalbuminuria is regarded as an early marker of DKD, it does not consistently predict the development of ESRD (*Roshan & Stanton, 2013*). The risk of developing ESRD within ten years in patients with macroalbuminuria is 9.3 times higher than in patients with

normo-albuminuria (*Berhane et al., 2011*). A meta-analysis of 28 cohorts with follow-up periods of up to 10 years showed that a 30% reduction in UACR lowers the risk of ESRD by 20–32%, particularly among patients exhibiting higher baseline UACR, where the intervention effect is more pronounced (*Heerspink et al., 2019*). These findings underscore that early identification and intervention in patients with macroalbuminuria can not only delay CKD progression but also significantly mitigate the risk of ESRD and associated complications.

Recently, novel drugs such as SGLT2 inhibitors and non-steroidal selective mineralocorticoid receptor antagonists have been proven to reduce albuminuria and protect the heart and kidneys in patients with diabetic nephropathy (*Heerspink et al., 2020*; *Jongs et al., 2021*; *Neuen et al., 2024*; *Wheeler et al., 2021*). Studies have shown that both SGLT2 inhibitors and MRAs can reduce albuminuria by 30% to 50% (*Agarwal et al., 2023*; *Epstein et al., 2006*; *Perkovic et al., 2019*). During treatment aimed at reducing albuminuria, improvements in the ECW/TBW may reflect reductions in fluid overload and stabilization of renal function, which could guide further adjustments in therapy. For patients with macroalbuminuria, continuous monitoring of the ECW/TBW is valuable for dynamically evaluating treatment efficacy and preventing further progression of kidney disease.

Additionally, the progression rates of DKD demonstrate significant variability among patients, influenced by factors such as genetic susceptibility, persistent hyperglycemia, and hypertension. Some patients may rapidly progress from normo-albuminuria to macroalbuminuria. Regular monitoring of the ECW/TBW provides an effective tool for individualized management, especially for high-risk patients with quickly progressing diseases. Tracking changes in the ECW/TBW can help adjust intervention measures promptly and optimize treatment strategies.

### Innovation

This study is innovative as it is the first to demonstrate the predictive ability of the ECW/TBW for albuminuria in T2DM patients across different CKD stages, particularly its ability to independently predict macroalbuminuria in the early stages of CKD. By integrating clinical indicators such as age, diabetes duration, and blood pressure, the prediction accuracy was significantly enhanced. Furthermore, this study validated the utility of the extracellular water ratio, determined by BIA, as a convenient and low-cost predictive tool for albuminuria, making it especially suitable for resource-limited settings. These findings provide new insights into the early screening and management of renal injury in T2DM patients.

### Limitations

Although this study provides a novel approach for the early screening of DKD, there are several limitations. Firstly, as a cross-sectional study, it does not establish a causal relationship. Secondly, while BIA offers the advantages of simplicity and high repeatability, its accuracy can be affected by the patient's condition (*e.g.*, oedema). Future studies should validate the causal relationship between the extracellular water ratio and albuminuria through prospective cohort studies and assess the utility of this indicator in different

T2DM subgroups. Additionally, although this study confirmed the predictive value of the extracellular water ratio in T2DM patients with better renal function, its role in advanced CKD patients remains unclear. Future research should explore how to integrate noninvasive tools like BIA with traditional indicators (*e.g.*, UACR and eGFR) to develop a multidimensional early screening model for T2DM patients with advanced renal impairment, optimizing the diagnosis and treatment of DKD. We have also taken an interest in the correlation between phase angle and chronic kidney disease. Regrettably, our InBody device does not measure this parameter. We anticipate that our forthcoming research will delve into the relationship between phase angle and diabetic nephropathy.

## CONCLUSION

This study demonstrated that ECW/TBW measured by BIA was significantly associated with albuminuria levels, especially macroalbuminuria, and showed predictive value for macroalbuminuria in patients with preserved renal function among Chinese T2DM patients. To ensure methodological rigor, strict quality control of measurements was implemented, multiple confounders were adjusted for, and internal validation of the predictive models was performed. Nevertheless, the cross-sectional design and single-center setting limit the generalizability of these findings. Future multicenter, longitudinal studies are warranted to further validate the clinical utility of this approach.

## ACKNOWLEDGEMENTS

The authors thank all the physicians and participants of the study for their cooperation and generous participation.

### Funding
This work was supported by the Natural Science Foundation of Shaanxi Province, China (Grant No. 2023-JC-YB-742), the Key Research and Development Project of Shaanxi Province (Grant No. 2023-ZDLSF-40), and the Innovation and Development Project of Tangdu Hospital, China (Grant No. 2021SHRC046; Grant No. 2021SHRC061; Grant No. 2021SHRC007). The funders had no role in study design, data collection and analysis, decision to publish, or preparation of the manuscript.

### Grant Disclosures
The following grant information was disclosed by the authors:
Natural Science Foundation of Shaanxi Province, China: 2023-JC-YB-742.
Key Research and Development Project of Shaanxi Province: 2023-ZDLSF-40.
Innovation and Development Project of Tangdu Hospital, China: 2021SHRC046; 2021SHRC061; 2021SHRC007.

### Competing Interests
The authors declare there are no competing interests.

## Author Contributions

- Aili Yang conceived and designed the experiments, performed the experiments, analyzed the data, prepared figures and/or tables, and approved the final draft.
- Xinwen Yu conceived and designed the experiments, performed the experiments, analyzed the data, prepared figures and/or tables, and approved the final draft.
- Zhiqiao Fan performed the experiments, analyzed the data, prepared figures and/or tables, and approved the final draft.
- Yuxin Jin performed the experiments, analyzed the data, prepared figures and/or tables, and approved the final draft.
- Fei Sun performed the experiments, analyzed the data, prepared figures and/or tables, and approved the final draft.
- Xin Wang performed the experiments, authored or reviewed drafts of the article, and approved the final draft.
- Xi Yuan performed the experiments, authored or reviewed drafts of the article, and approved the final draft.
- Langlang Liu performed the experiments, authored or reviewed drafts of the article, and approved the final draft.
- Guohong Zhao conceived and designed the experiments, authored or reviewed drafts of the article, and approved the final draft.
- Bin Gao conceived and designed the experiments, authored or reviewed drafts of the article, and approved the final draft.

## Human Ethics

The following information was supplied relating to ethical approvals (i.e., approving body and any reference numbers):

The Institutional Review Board of the Medical Ethics Committee of the Tangdu Hospital, the Fourth Military Medical University (No. K202207-05).

## Data Availability

The raw measurements are available in the Supplementary File.

## Supplemental Information

Supplemental information for this article can be found online at http://dx.doi.org/10.7717/peerj.19780#supplemental-information.

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
