# Peer review of "The association between the extracellular water-to-total body water ratio and albuminuria in Chinese type 2 diabetes mellitus patients"

_PeerJ, doi:10.7717/peerj.19780_

## Round 0.1 · original submission · Major Revisions

**Language Note:** The review process has identified that the English language must be improved. PeerJ can provide language editing services - please contact us at [email protected] for pricing (be sure to provide your manuscript number and title). Alternatively, you should make your own arrangements to improve the language quality and provide details in your response letter. – PeerJ Staff

·

Basic reporting

More literature references are needed to cite the nexus between impedance analysis and biochemical considerations. The quality of the language used offers scope for further improvement

Experimental design

The methods did not specify the gender and age of the study subjects included

Whereas de Souza Francisco, D et al in their paper, BMC Sports Sci Med Rehabil 16, 34 (2024). https://doi.org/10.1186/s13102-024-00825-5, have indicated that phase angle is an important determinant with reference to all-cause mortality in CKD. Why did the authors not consider this (Phase angle)?

Various factors such as Adipose Tissue, Body Composition, Fluid intake, Type of food, and menstrual cycle affect the impedance analysis. Did the authors take special care while recruiting the study subjects?

The authors have mentioned in the Exclusion criteria that they had excluded subjects based on (1) incomplete data, (2) presence of other kidney diseases, and (3) severe liver disease. Were there subjects with muscle disorders, and if so, were they excluded?

A publication by Janis M Dionne et al https://doi.org/10.1161/HYPERTENSIONAHA.120.1669 has highlighted the role of mean arterial pressure in Chronic kidney disease progression.
Did the authors of the present manuscript consider this at all? Measurement of systolic and diastolic blood pressure has its inherent limitations.

The authors have mentioned under limitations:- “Secondly, while BIA offers the advantages of simplicity and high repeatability, its accuracy can be affected by the patient’s condition (e.g., oedema)”. The authors could have done lab investigations such as CBC, serum electrolytes, LFT, and Total protein, Albumin, which could have provided insight into the aspect of oedema.

Insulin resistance is a major determinant of chronic kidney disease. Why did the authors not take cognizance of the same?

A recent publication, namely “Association between triglyceride-glucose index and chronic kidney disease: results from NHANES 1999–2020,” published in Int Urol Nephrol. 2024 Jun 10;56(11):3605–3616. doi: 10.1007/s11255-024-04103-8, has indicated the role of Triglyceride Glucose index(TyG) in CKD. Had the authors included this component, namely TyG, in their study, it would have fortified the results obtained from the impedance analyser

Validity of the findings

The conclusions could have been more robust had the authors used their kind discretion in enhancing the validity of the methods.

Additional comments

The study is an earnest attempt that needs to be complemented, as CKD is an important consequence of Type 2 diabetes mellitus, and the authors have used impedance analysis to document changes in the body water as a function of the biomarkers. However, had a few more important biochemical analytes been included, the validity of the endeavor would have been stronger still

Reviewer 2 ·

Basic reporting

The language needs improvement in a few areas. The comments have been included in the PDF attached.

Experimental design

The research is novel in exploring a newer marker using BIA in predicting albuminuria, in other words, the progression from incipient nephropathy to overt nephropathy in T2DM. Limitations may include the generalizability of the ECW/TBW as ethnicity differences are known to exist (Davenport A, Hussain Sayed R, Fan S. The effect of racial origin on total body water volume in peritoneal dialysis patients. Clin J Am Soc Nephrol. 2011 Oct;6(10):2492-8. doi: 10.2215/CJN.04130511).

Validity of the findings

-

Additional comments

There are redundancies noticed: lines 213 and 225 are the same

Annotated reviews are not available for download in order to protect the identity of reviewers who chose to remain anonymous.

·

Basic reporting

Clear and unambiguous, professional English used throughout.

Literature references, sufficient field background/context provided.

Professional article structure, figures, and tables. Raw data shared.

Self-contained with relevant results to hypotheses.

Experimental design

Original primary research within the Aims and Scope of the journal.

Research question well defined, relevant & meaningful. It is stated how research fills an identified knowledge gap.

Rigorous investigation performed to a high technical & ethical standard.

Methods are described with sufficient detail & information to replicate.

Validity of the findings

Meaningful replication encouraged where rationale & benefit to literature is clearly stated.

All underlying data have been provided; they are robust, statistically sound, & controlled.

Conclusions are well stated, linked to the original research question & limited to supporting results.

---

## Round 0.2 · accepted · Accept

Dear Dr. Gao,

Thank you for submitting the revised version of your manuscript. After a thorough evaluation of your revisions by Reviewers and me, I am pleased to inform you that all reviewer comments have been satisfactorily addressed. Accordingly, your manuscript is now accepted for publication in PeerJ.

Sincerely,
Stefano Menini

·

Basic reporting

The authors have taken cognizance of the suggestions/recommendations made by this reviewer and implemented the same satisfactorily.

Experimental design

Looks more robust and objective, in comparison to the preliminary version of this manuscript.

Validity of the findings

The validity of the findings has been enhanced, following the Reviewer Comments.

Additional comments

The manuscript looks more well rounded and comprehensive. Deemed fit for Publication.

·

Basic reporting

Clear and unambiguous, professional English used throughout.

Experimental design

Original primary research within Aims and Scope of the journal.

Validity of the findings

Impact and novelty not assessed. Meaningful replication encouraged where rationale & benefit to literature is clearly stated